# Cancer as a Chronic Illness in Colombia: A Normative Consensus Approach to Improving Healthcare Services for those Living with and beyond Cancer and Its Treatment

**DOI:** 10.3390/healthcare9121655

**Published:** 2021-11-29

**Authors:** Cindy V. Mendieta, Maria Elizabeth Gómez-Neva, Laura Victoria Rivera-Amézquita, Esther de Vries, Martha Lucía Arévalo-Reyez, Santiago Rodriguez-Ariza, Carlos J. Castro E, Sara Faithfull

**Affiliations:** 1Department of Clinical Epidemiology and Biostatistics, Pontificia Universidad Javeriana, Bogotá 110231, Colombia; estherdevries@javeriana.edu.co; 2Clinical Nursing Department, Faculty of Nursing, Pontificia Universidad Javeriana, Bogotá 110221, Colombia; m.gomezn@javeriana.edu.co; 3Grupo de Investigación en Ciencias de la Rehabilitación, Escuela de Medicina y Ciencias de la Salud, Universidad del Rosario, Bogotá 111711, Colombia; laurav.rivera@urosario.edu.co; 4Patient and Caregiver, Bogotá 111711, Colombia; marthalucia75@yahoo.com; 5Grupo de Bioquímica Experimental y Computacional, Facultad de Ciencias, Pontificia Universidad Javeriana, Bogotá 110221, Colombia; santiago_rodriguez@javeriana.edu.co; 6Scientific Director of Liga Colombiana Contra el Cáncer, Bogotá 110231, Colombia; cjcastroe@gmail.com; 7School of Health Sciences, Faculty of Health and Medical Sciences, University of Surrey, Guildford GU2 7XH, UK; sara.faithfull.prof@gmail.com

**Keywords:** living with and beyond cancer, late effects, survivorship, nominal consensus, chronic illness, cancer health services

## Abstract

Cancer survivorship care in Colombia is of increasing importance. International survivorship initiatives and studies show that continuing symptoms, psychological distress, and late effects impact the quality of life for survivors. Priorities for quality survivorship according to Colombian patients and clinicians are unknown. We undertook a nominal consensus approach with 24 participants using virtual meeting technology to identify the priorities for cancer survivorship. We applied an iterative approach conducted over eight weeks with five workshops and one patient focus group followed by a priority setting survey. The consensus group established six main themes, which were subsequently evaluated by experts: (i) symptoms and secondary effects of cancer; (ii) care coordination to increase patient access and integration of cancer care; (iii) psychosocial support after cancer treatment; (iv) mapping information resources and available support services for long-term cancer care; (v) identifying socioeconomic and regional inequalities in cancer survival to improve care and outcomes; and (vi) health promotion and encouraging lifestyle change. The order of priorities differed between clinicians and patients: patients mentioned psychosocial support as the number one priority, and clinicians prioritized symptoms and surveillance for cancer recurrence. Developing survivorship care needs consideration of both views, including barriers such as access to services and socioeconomic disparities.

## 1. Introduction

In 2020, around 19.3 million persons were newly diagnosed with cancer globally, a number expected to rise to 28.9 million by 2040 [1]. In Colombia, cancer is the third most frequent cause of mortality, and incidence is expected to increase by 86.5% by 2040 compared to 2020 [1]. The Colombian health system is public-private with universal and mandatory coverage and is divided into three types of insurance. The subsidized regime (state-financed) covers people unable to pay for health care, the special regime is allocated by occupation, for members of the public workforce, and the contributory regime (third payer) represents the country’s main workforce [2]. Differences in cancer survival have been described in Colombia, with rural and more vulnerable populations being at a disadvantage, as mainly attributed to socioeconomic inequalities and access to care [3,4]. These lower-income and rural populations are mostly affiliated with the subsidized regimes [3,4,5,6]. Less than 50% of new breast, colorectal, and stomach cancer patients are diagnosed in early stages, meaning cancer patients need more supportive care for this advanced disease [2]. These later stage patients are more likely to be in the subsidized regime [7].

Despite the financial and social challenges, cancer survival in Colombia has increased for most cancer types, although survival rates remain lower compared to high-income countries [4]. This improved survival has resulted in a growing number of people now living with and beyond cancer diagnosis and treatment. They need support, such as surveillance for recurrence, long-term symptom management, psychosocial care and health promotion [4], and recognizing cancer now as a long-term or chronic condition [1,2,3,4,8].

The United States, one of the first countries to define cancer survivorship and to recognize the need for long-term cancer care, describe a cancer survivor as a person with cancer, who from the moment of their initial diagnosis, treatment, until their death, needs care [9]. In the United Kingdom (UK) several guidelines, policies, and supporting tool kits for living with and beyond cancer have been developed to address cancer survivorship [10] aiming to improve the long-term outcomes for cancer patients. Despite these policy initiatives, there is still a wide disparity in the survivorship services that people with cancer can access globally.

In Colombia, cancer survivorship refers to a patient alive after a determined number of years, depending on the cancer type [11] with little focus on the health care plan for cancer as a chronic illness. According to the Colombian Cancer Control Plan 2012–2021, survivorship includes “*physical, psychosocial and economic issues related to cancer, from diagnosis to the end of life*” [12]. This includes access to healthcare, treatment (short- and long-term secondary effects), surveillance for the occurrence of second primary cancers and recurrences, and quality of life [12]. However, the current healthcare system does not provide services specifically to meet survivors’ needs, and only a few institutions offer a program for cancer patients after their discharge from active treatment and return to “normal” life [11]. A recent narrative review of cancer survivorship in Colombia proposes four general recommendations: (i) recognizing cancer survivorship as a distinct phase of cancer, (ii) methods for tracking and metrics for survivorship, (iii) assessing and monitoring cancer symptoms and quality of life of survivors and (iv) publishing evidence-based guidelines to meet the specific population needs of Colombians with cancer [11]. Building on this narrative review, we wanted to understand the opinions and experiences of those providing and receiving cancer care within Colombia. We explored gaps in service provision for long-term cancer management, symptoms and areas of supportive care need, barriers to using existing health services, health promotion after cancer treatment, and novel support solutions. Our aim was to establish consensus statements and priorities to inform future practice development and policy in cancer survivorship within Colombia.

## 2. Methods

We used a qualitative consensus group approach using an adapted nominal group technique (NGT) [13], a method traditionally conducted face-to-face and used when there is a paucity of evidence in the area [14]. We applied an iterative approach conducted over eight weeks with five workshops and with experts in Colombia invited via an announcement in the Living With and Beyond Cancer (LWBC) clinical network. Those who accepted were allocated to distinct working groups according to their experience. Research and interests from these seven experts were assigned to the cancer survivorship workshops.

The workshops were moderated by the research facilitators in the UK and Colombia and each of them began with a brief presentation summarizing recent international and Colombian literature for the topics to be discussed in that session. The pre-established topics were: (i) mapping information resources for patients on secondary consequences of cancer treatment, (ii) rural medical care, and (iii) digital health interventions. Additional topics were able to be added by the experts.

The workshops were conducted by one of the workshop members selected according to their expertise in the field, followed by a discussion online with a duration of approximately 2 h. Each workshop had the participation of the same inter-disciplinary group of experts which included an: oncologist (1), physician/clinician in rural medicine (1) nurse (2), physiotherapist (1), nutritionist and dietitian (1) and patient and caregiver representative (1) with an average attendance at workshops of 86%. After each meeting, we recorded the workshop and organized the discussion in minutes, with a pre-established format. It included the contributions of each researcher and a review in English and Spanish by the chief researchers and was agreed by the experts at the next workshop. Researchers performed an analytic reading of the minutes and transcripts to develop themes to construct consensus statements, which were then agreed upon with the group at the final workshop.

From these workshop themes we established a series of questions to ask patients about their experience of survivorship (see Appendix A Appendix A. Focus group questions English and Spanish) and we conducted a focus group with survivors to contribute to the themes already raised by our expert panel. The participants of the focus group were patients already known by the experts in their clinical setting. Participants received a formal invitation and completed an informed consent form which included permission for the recording and publication. Those who confirmed their participation and completed consent were scheduled in a group meeting using Microsoft Teams. The focus group was recorded via audio, due to COVID-19, and we used virtual meetings rather than face-to-face. The research assistant transcribed the verbal and non-verbal communication and translated it into English and conducted a thematic analysis (Appendix A Appendix A).

The main survivorship themes were identified during workshops and the focus group and discussed later in a final meeting with the workshop’s experts. All themes were worded as priorities that were used as the basis for the prioritization survey (in Spanish) with a duration of approximately 10 minutes which was sent by email to the broader groups of healthcare providers, patients and survivors who participated in the Colombian network of LWBC. This included: a physician (2), specialists in family medicine (2), a palliative care physician (1), cancer researchers (9), allied health professionals (3), patients and cancer survivors (4), psychiatrist and psychologist (2), and medical auditor (1). They had the possibility to suggest new priorities through an open question survey (see Appendix A Appendix A). The themes previously identified by the expert panel were organized according to the prioritization order by the participants and an assignment for each category in the first and second order (Table 1).

## 3. Results

As a result of discussion at the initial workshop, the experts identified additional themes to be discussed making six topic areas including; (i) mapping information resources for patients on secondary consequences of cancer treatment, (ii) rural medical care in cancer, (iii) exploring digital health interventions, (iv) potential interventions that could benefit individuals living with and beyond cancer and (v) health promotion and (vi) lifestyle interventions.

During the first workshop, experts expressed the need to organize a focus group with patients to explore their experiences with the illness, treatment, side effects, access to information, difficulties of the health system, gaps, and unmet needs (Appendix A Appendix A) so that this could inform the priorities. The focus group was conducted with four adult participants, three females, one diagnosed with non-Hodgkin’s lymphoma, two with breast cancer and an older adult man with colon cancer. Two of the participants in the focus group were residents of rural areas and all had received multimodality cancer therapies. The participants all contributed to the contributive regime but were affiliated to distinct Entidades Promotoras de Salud (EPS—health insurance) and were attended to in different Institutes of Health Attention (IPS—clinics and hospitals). In the following sections we combine the results of the workshops with those of the focus group to explore the themes.

### 3.1. Symptoms and Secondary Effects of Cancer Treatment, Identifying Those at Higher Risk, and Addressing Symptom Management

Participants from the focus group identified a wide range of problematic symptoms after cancer treatment (fatigue, nausea, pain osteoporosis, and peripheral neuropathy) and were uncertain if these were related to treatment or should be symptoms of concern. Some of these were transient whilst other symptoms continued after cancer treatment was completed, impacting on their ability to work and quality of life. Patients described a lack of focus by their medical teams on chronic symptoms, such as treatment-related pain. One participant described “*There have been hard time when the pain for the first time bent me over and made me cry physically*” P3.

Patients also described that they used complementary therapies to manage side effects of treatment but rarely disclosed this to their medical teams. Clinicians focused on the need for effective acute symptom management and felt they lacked knowledge for management of chronic symptoms such as fatigue. Non-pharmacological interventions for chronic symptoms as an adjunct to pharmacological management of side effects such as rehabilitation, physical activity, self-management or acupuncture were not mentioned.

### 3.2. Care Co-Ordination to Increase Patient Access and Integration of Cancer Care

Patients and clinicians alike recognized care coordination as a major issue to access health care professionals. One patient in the focus group described the multiple clinicians they were referred to in addressing their side effects from cancer treatment “*I was seen by the oncologist…the dermatologist…the rheumatologist…they do not see us as sick patients but as human beings who need to be rescued*” P1. The pathways back into cancer care for survivors were often uncertain or delayed and required multiple visits to institutions rather than an integrated approach to care. Clinicians described that this was made worse by the inability to share patient records or data across systems. A lack of data on many of the cancer survivors, side effects or metrics for use of existing services were seen as issues. Clinicians mentioned the relationship with primary care as being important, but they were not very confident in primary care physicians’ knowledge of cancer. Integrated care was seen by the clinicians as a framework for enhancing care quality through better integration of health services, including strategies to promote access to palliative care, standardized care pathways, and multidisciplinary teams. Clinicians identified newly installed telehealth as a potential tool for remote service provision and a future strategy for providing care coordination and access to specialist advice.

### 3.3. Providing Psychosocial Support after Cancer Treatment

The need for psychological support was considered a priority by both clinicians and patients. Patients described medical staff as being mainly focused on treatment and management but had good relationships with their medical teams. One patient described the emotional distance experienced when discussing concerns with their family “*There is a really disparity with the people of your family…you do not feel comfortable that your family is (thinking) poor you… because you do not want them to see you as a sick person*” P4.

Clinicians in the workshop described the need for a screening tool to help them identify psychological distress in Spanish to help them identify those with needs and which can be used routinely.

### 3.4. Mapping Information Resources and Available Support Services for Managing Long-Term Cancer Care

Information provision and self-management were described as a priority for managing the consequences of cancer treatment by both health care professionals and patients. Patient in the focus group experienced a lack of information and mentioned receiving conflicting advice especially concerning nutrition and exercise. One participant described “*There is a lot of misinformation and (the) people in the middle of that misinformation and the remoteness of the doctors… you don’t know who to call, everyone begins to have an opinion, everyone lost, anguished*” P4. Patients described cancer information as being focused on short term information and that there was a gap in exploring long term treatment, potential side effects, as well as recovery plans.

### 3.5. Identifying Regional Differences and Socioeconomic Inequalities in Cancer Survival to Improve Cancer Care and Outcomes in Vulnerable Populations

Clinicians felt that optimal cancer treatment was compromised for patients living in rural areas, factors such as the distance to the cities where they could receive treatment and the lack of oncological knowledge in rural areas. The lack of protocols for integrated cancer services in rural areas was seen as a priority for improving cancer care. Clinicians and patients understood the need for centralization of highly specialized cancer services in the country but highlighted the urgency to develop a more equitable survivorship cancer care across all regions. One participant in the focus group described the high cost and burden of travel “*To cure me every third day I had to travel from my home to the clinic…I would have been spending between one and a half to two hours travelling. So just imagine the route I had to take I was surely going to die*” P2. Funding for such travel and disability payments from insurance providers was described by patients as a fight and created socioeconomic inequalities.

### 3.6. Health Promotion and Encouraging Lifestyle Change

None of the participants in the focus group had access to nutritional support, physiotherapist or occupational therapists. Patients expressed concerns about the wide variety of available resources on diet information and the confusing messages they present “*…Food is a theme that does not tell you anything and information is reaching you everywhere*” P4.

Clinicians described a cancer diagnosis as a potential opportunity to modify unhealthy behaviours such as increasing physical exercise, dietary change, or smoking cessation especially in reducing cancer recurrence.

### 3.7. Prioritization of Themes

Results of the prioritization survey are described in Table 1. Health care professionals in the wider LWBC network and patients agreed on the top 4 priorities. Whereas patients prioritized psychosocial support, healthcare professionals prioritized management of symptoms and secondary effects of cancer and identifying those patients at highest risk. Both groups identified that care coordination was a major priority. Moreover, healthcare professionals suggested to include “humanization of care” in the comprehensive route for health care attention for cancer, a document issued by the ministry of health. They also stressed the importance of implementing a survivorship pathway for those with cancer as a chronic illness.

## 4. Discussion

This consensus group provided an opportunity to explore priorities for cancer survivorship care in Colombia from different perspectives. Previous research recommended to improve early cancer diagnosis to increase survivorship and focus on long-term survivorship care. The Colombian Ten-Year Plan for Cancer Control mentions improving the quality of life of cancer patients and survivors as an important strategic aim without mentioning how to achieve this [12]. Franco Rocha et al. [11] identified in their narrative review a vision of cancer survivorship care mirroring that of the USA, although, despite a decade of implementation of survivorship policy, there remain significant barriers and inequity to survivorship care provision in North America [15]. This narrative review lacked specificity of areas of action needed; therefore, our results are a useful contribution to future strategic planning.

In Colombia, the social determinants of health differ from those in North America for its social inequalities, different health insurance model, centralization of health services, and geographic barriers that make even access to primary care difficult (especially for those who reside in scattered rural areas). The lack of individualization of care packages limits options for personalized care. There is a limited offer of rehabilitation, and palliative care and pain control with most health professionals not trained in palliative care and the offer of palliative care services mainly limited to specialist centers in large cities [16,17].

The use of complementary therapies by patients to alleviate symptoms and for pain control was not discussed with medical teams, and this has been well described in the literature with it estimated that over 40% of patients report using complementary therapies since cancer diagnosis [18]. Although there is evidence for benefit for some cancer symptoms especially mind-body interventions and acupuncture, there is potential unwanted drug-drug interactions with some herbal remedies [19], indicating the importance of explicitly exploring this topic during consultations.

Patients and clinicians recognized different priorities for survivorship care, both agreed on the importance of improving cancer pain management and psychosocial support, but patients ranked psychosocial support as most important, whereas clinicians prioritized treatment of symptoms and side effects of cancer therapy. The diversity and significance of cancer symptoms makes it understandable that clinical teams prioritize symptom management, but it is also important to recognize the subjective patient experience to inform clinical practice [20]. Psychological distress is recognized as a side-effect of cancer and its treatment, with reports of moderate levels of anxiety in 4–50% of cancer survivors [21,22]. Depression and anxiety are often underdiagnosed and under-treated among cancer patients [23]. The dominant focus of current cancer survivorship care is disease surveillance (e.g., recurrence monitoring, addressing physical late effects). The need for psychosocial care was identified as unmet and considered a priority for establish assessment and clinical services.

Coordination of care to increase access to services and integration of a heavily fragmented health system was seen as priority number two for both clinicians and patients. Multiple referrals to other healthcare providers, frequently in different institutions, without communication between the providers left patients without a central spokesperson in the system. The lack of data on cancer survivors, or metrics on where they are in the system, was seen as key to improving existing services [7]. Data could be improved by establishing data sharing of cancer and non-cancer follow-up across healthcare settings and specialties [12]. Survivorship care plans were not provided for any of our patients in the focus group and this is not unique to Colombia [15]. Survivorship care plans are often seen as a mechanism to improve care coordination and patient support, but a recent systematic review and meta-analysis found that, although feasible, there was no evidence that they improved patient outcomes [24]. Overcrowding in surveillance clinics and constraints in the treatment-focused setting have driven change in the UK with risk-stratified pathways and new models of survivorship care [25], but this change is dependent on patient data and planning future projections for capacity and service utilization.

Clinical practice guidelines are available in Colombia, but the guidance for palliative cancer care does not include nutrition, psychotherapy, and invasive pain therapies [26]. All these therapies are covered by the health system, but their access is limited according to diagnosis, pathology, and health insurance [7]. The lack of access to rehabilitation for nutritional advice and exercise-related interventions is often suboptimal, but can provide improvement symptom management, physical function and independence [27,28]. The European Society for Clinical Nutrition and Metabolism proposes screening tests and nutrition therapy in the early stages of cancer treatment to prevent malnutrition and its consequences [29], and systematic reviews have shown that exercise decreases the adverse effects of cancer treatment, reduces recurrence and mortality for many cancers [30,31,32]. Patients with nutritional risks have a 2–4-fold higher risk of complications, length of hospital stay, and healthcare-related costs [33,34]. Yet, in 2019, only 18% and 24% of the patients with colorectum and stomach cancer, respectively, received a nutritional assessment [7] and the prevalence of malnutrition in cancer is closer to 20 to 70% [35].

Access to information was seen as a 4th in the priorities as it influences communication and an individual’s decision-making [36]. There is a need for interventions such as adapting communication, information provision, and cancer care coordination to improve the quality of cancer care [37]. For instance, patients with low levels of literacy often have poorer health outcomes and poor access to information [38]. New methods to accessing information have emerged during the COVID-19 outbreak in Colombia, which has resulted in online and e-health programs, facilitating remote access to, which has the potential for expansion to facilitate the communication between clinicians and cancer patients to reduce health access barriers [39].

Like many countries, there are regional differences and significant barriers for those who live outside of the main cities. The huge geographical disparities in degrees of development, quality of roads, availability of specialized medical care, and income inequalities affect cancer health attention [7]. Unsurprisingly, cancer mortality is higher in remote rural areas and among the socio-economically more disadvantaged populations [3,37]. An international meta-analysis shows that people from remote and rural areas are 5% less likely to survive their cancer and are more likely to have the advanced disease [40]. They systematically present at a later stage of diagnosis and have longer times between initial symptoms, diagnosis, and treatment [3,37]. Additionally, research shows that patients with longer traveling times to healthcare facilities have poorer outcomes [41], and that this subsequently impacts financially on their families [42]. Strategies such as improving primary care cancer education, outreach allied health support and nurse navigators to help move across localities have worked in other healthcare systems [43], but structural factors that underlie access issues such as poverty, rural or remote location and low socioeconomic status remain. Such social determinants of health need to be considered when designing future survivorship models of care.

The strength of this work was that we included patient voices in the discussion and consensus alongside those of physicians, nurses, and allied health professionals, conducting the workshops in both Spanish and English to be as inclusive as possible. The nominal group technique usually does not include patient perspectives, but the focus group helped our understanding and enriched the findings. It is important in developing systems of care to include patient voices as we yet do not have a good understanding of how cancer survivors’ access and use health care resources in Colombia. Limitations of having only one patient focus group with no representatives from wider populations such as the full range of cancers, indigenous or scattered rural populations is recognized. We therefore probably did not reach saturation in relation to patient views yet managed to identify several shared important topics. With the clinician themes we were able to survey a different group of clinicians which helped confirm our themes and develop the priorities. In future research, a Delphi technique with multiple iterations of questions for a wider population would be stronger in developing consensus, but also a patient experience survey to identify issues and needs after cancer treatment and a larger qualitative study would be useful.

This nominal group technique enabled us to explore priorities for improving survivorship care in Colombia considering both patient and diverse clinical perspectives. We identified six specific priorities to inform health care providers and policymakers on improving cancer survivorship pathways with the aim to improve access to services and the quality of life of patients who live with and beyond a cancer diagnosis. An important next step to investigate is to identify how cancer centers could, in close collaboration with the health insurers, develop survivorship pathways considering the priorities raised. This nominal consensus is a small step into developing greater recognition of the chronic care needs and priorities for those living with and beyond a cancer diagnosis in Colombia.

## Figures and Tables

**Table 1 healthcare-09-01655-t001:** Priorities of healthcare professionals and patients/survivors in Colombia.

Priority	Health Care Providers (*n* = 20)		Patients/Survivors Only (*n* = 4)	
	Order Number (Total Score)	Times Mentioned as 1–2	Order Number (Total Score)	Times Mentioned as 1–2
Explain expected symptoms and secondary effects of cancer to the patients, identifying those for which the patient is at higher risk	1 (49)	11	3 (14)	1
Care coordination to increase patient access and integration of cancer care	2 (58)	10	2 (13)	2
Providing psychosocial support before, during, and after cancer treatment	3 (72)	7	1 (6)	2
Provide a list of information resources and available support services for managing long-term cancer care to patients and caregivers	4 (75)	4	4 (15)	2
Identifying regional and socioeconomic differences inequalities in cancer survival and formulate plans to improve cancer care with a differential focus for vulnerable populations	5 (77)	6	6 (15)	0
Health promotion and encouraging lifestyle changes for patients and survivors	6 (83)	2	5 (15)	1

## Data Availability

Not applicable.

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
