# Peer review of "Cancer as a Chronic Illness in Colombia: A Normative Consensus Approach to Improving Healthcare Services for those Living with and beyond Cancer and Its Treatment"

_healthcare, 2021, doi:10.3390/healthcare9121655_

Round 1
Reviewer 1 Report
I have reviewed the manuscript titled, “Cancer as a Chronic Illness in Colombia: a Normative Consensus Approach to Improving Healthcare Services for Those Living with and Beyond Cancer and its Treatment,” for the second time now.
There appears to be improvement in the manuscript after the authors addressed some of the comments and concerns raised by this reviewer. Below are my comments and suggestions on the narrative of the Results Section some of which may not have been included during my previous review.
According to the international Committee of Medical Journal Editors ( http://www.icmje.org/recommendations/browse/manuscript-preparation/preparing-for-submission.html), there are certain principles and practices that need to be adhered to in writing for a medical/ healthcare journal unless the author guidelines specific to that journal state otherwise. In this manuscript, the Results Section is rampant with the authors’ opinions, comments, deductions, assumptions, inferences, and references (References 15 to 37).
The Results Sections MUST be FREE from the authors’ opinions, comments, claims, inferences, and references. It should not be buttressed with sourced or unsourced material to make it look better or appear evidence-based. Because this is a qualitative study, it may consist of the subjective or objective narrative of interview material (opinion, comments, statements, etc.) obtained directly from study participants or focus groups.
Therefore, this reviewer recommends a total rewrite-up of the Results Section, by removing all the authors’ opinions, comments, deductions, assumptions, inferences, and references (References 15 to 37) and replacing theses with unqualified descriptive narrative as obtained directly from study participants and focus groups.
Reviewer 2 Report
Thanks for complying with my suggestions. I do not have any further comment. I suggest publication.
Reviewer 3 Report
This manuscript reports the results of a nominal group process to identify priorities for cancer survivorship care in Colombia. Given the lack of country specific research and policy in this area to date, this study provides valuable data that has the potential to inform healthcare policy and potentially improve care. Overall I found the paper to be mostly well written and logical in its approach. I have noted a number of issues below which I would encourage the authors to reflect upon and which may improve the manuscript.
Major comments:
Methods
More information about the workshops is required.
Were the workshops also delivered online or face to face? This is not clear
Did all workshops follow the same format (eg lit review, then discussion) or was this only for the first workshop? What was the focus of the lit review/s and who determined this? Did the same group of experts attend all 5 workshops? How were these experts identified and selected? How were the consensus statements shared with the experts and when did this occur? Did they have an opportunity to give feedback/change the statements?
How were the participants in the prioritization survey recruited and how was the survey administered? Also inaccurate to state that it was sent to a different group of clinicians (this ignores the fact it was also sent to patients/survivors).
Results
While the results section was well described, I think it would be clearer to the reader if it was presented in chronological order ie - I would present the information about the development of the consensus statements first (ie information on pages 6 and 7), then the prioritization survey results last.
I think Table 2 could be excluded from the main manuscript and just referred to in the supplementary material.
For Table 1, I'm not sure why the views of the patients (who are also seperated out into their own group) are included/grouped with the clinicians. Makes more sense to have just a clinician group (n=20) and seperate patient group. I also don't know what the "total score" column refers to - is this critical to include?
Minor comments:
page 1, line 38 - third "highest" cause (missing word)
page 1, lines 40-43 - I found the description of the healthcare system difficult to understand, consider rewording to improve clarity.
page 2, line 72-73 - remove one of the "specifically"
page 8, line 243, saying cancer mortality is higher is the same as survival lower - dont need both
Focus group comments - I see EPS is defined in the notes, but would be helpful to explain what this is for non Spanish speakers.
Round 2
Reviewer 1 Report
The authors appear to have addressed this reviewer's feedback, comments, and concerns. Further review, if necessary, is left to the authors' discretion.
This manuscript is a resubmission of an earlier submission. The following is a list of the peer review reports and author responses from that submission.
Round 1
Reviewer 1 Report
The Authors report on a nominal consensus approach to identify the priorities for cancer survivorship in Colombia. They used an approach base on a qualitative consensus group using an adapted nominal group technique. They applied an iterative approach conducted over 8 weeks with 5 workshops and one patient focus group followed by a priority setting survey. The article is of interest, since the topic is important and qualitative research is useful to investigate causal inference rather than statistical correlation, particularly in case of paucity of data. I have comments:
- Can you provide details on the structure of the workshops and the patient focus group discussion? In case it won’t be possible to do that with brevity, please add this information in the Supplementary material
- How were the six main themes selected in the workshops? How was the rate done? By frequency? By scoring relevance? Other?
- Can you please add in the supplementary material, the text of the survey used for prioritization?
- How did you ensure data saturation during the iterative process? Did you use, for example, a final focus group or workshop with a different population to be employed as external validation to check for data saturation? If not, please add this point into the limitations of the study.
- Was Quality of life explored as an item important for patient? I see no mention about QoL. Was it included in assessment of symptoms and secondary effect of cancer?
- What is the next point after this nice study? How would you tackle the issues arosen? What is the action plan? Please elaborate in the discussion section.
Reviewer 2 Report
I hope all the authors are safe and doing well.
This manuscript showed that the priorities of cancer patients(survivors) were different from those of clinicians who participated in the designed workshops.
MAJOR COMMENTS
My biggest concern regarding this paper are the followings.
1) The number of cancer patients are so small.
It seems a total of four cancer patients (survivors) had participated in the workshops. This manuscript does not describe how they were recruited -more explanation regarding the enrollment process is required than a voluntary participation. In addition, characteristics of these patients were hardly found in the footnote. The main text should describe about what types of cancer they had, how old were they, and their gender ... etc.
The biggest issue is that the opinions of these four patients included in the study cannot be generalized. The reason why patients prioritized psychosocial support more than other themes might be due to the fact that most of the patients were females. The disease severity, recurrence, and types of treatment they had might also have affected their thoughts as well.
2) The experts included a variety of healthcare professionals.
I couldn't find any information about how many clinicians, nurses, allied health professionals were included in the study.
what are the characteristics of the experts included in the study?
Is there any reason why you included a variety of healthcare professionals to the study? Why not just focus on clinicians/physicians?
3) The ethical statement did not include any information regarding whether this study was approved by an institutional review board.
4) As far as I could understand, five workshops (each about two hours) were held. I was wondering about the number of participants included in the workshop. Did they all participate in every sessions?
Moreover, why did you design five different sessions over eight weeks of period?
If all subjects had participated in every sessions, then their thoughts/opinion regarding the main issue could be affected by thoughts/opinions of others.
5) I think the introduction should focus more on why it is important to gather thoughts of cancer survivors and experts in the related area. My recommendation is to describe current incidence and mortality of cancer patients in Columbia in detail and to discuss about medical practices and political policies associated with cancer survivorship. It might be meaningful to examine types of cancer with higher survival rates as well.
6) How did you come up with the scores in table 1?
MINOR COMMENTS
- '(200)' in the abstract should be removed and so does (3181) in the introduction.
Reviewer 3 Report
1.Title: "Cancer as a chronic illness in Colombia: a normative consensus approach to improving survivorship and healthcare services for those living with and beyond cancer and its treatment." The title introduces a preconceived and rather unsubstantiated notion by the authors that some sort of consensus improves cancer survivorship. This reviewer contends that consensus can be reached to improve health care services but cancer mortality is not a matter of consensus, and as such the title needs to be rephrased to: “Cancer as a chronic illness in Colombia: a normative consensus approach to improving healthcare services for those living with and beyond cancer and its treatment.”
2. Method- This study which appears to be a qualitative design does not have a clear research question; the number and the composition of the focus groups (demographic variables, professional titles, etc.); the nature of the study area (urban vs rural, one center/health care facility vs multi-center study) is not clear, and how the qualitative data was analyzed is not clearly stated.
3. Although access to care is mentioned, it is interesting that the themes do not include the quality of care and the cost/affordability of services provided to cancer patients across the continuum of care (from screening all the way to terminal/palliative care).
4. The study falls short of recognizing the diverse nature of cancer diagnoses and appears to endorse a blanket consensus approach to all cancer patients.
5. This reviewer believes survivorship priorities are different for each individual cancer. Thus survivorship priorities for breast cancer are different from survivorship priorities for lung cancer, colorectal cancer, etc.
5. In sum, this study is a narrative of subjective opinions which do not amount to evidence-base/ new knowledge that can used in clinical or public health interventions that will improve cancer survivorship.